# Characterization of *Lactococcus lactis* 11/19-B1 Isolated from Kiwi Fruit as a Potential Probiotic and Paraprobiotic

**DOI:** 10.3390/microorganisms11122949

**Published:** 2023-12-09

**Authors:** Ken Ishioka, Nozomu Miyazaki, Kyoko Nishiyama, Tatsuo Suzutani

**Affiliations:** Department of Microbiology, School of Medicine, Fukushima Medical University, Fukushima 960-1295, Japan; m-nozomu@fmu.ac.jp (N.M.); kyoko@fmu.ac.jp (K.N.); suzutani@fmu.ac.jp (T.S.)

**Keywords:** *Lactococcus lactis*, probiotics, paraprobiotics

## Abstract

Probiotics are live bacteria used as food additives that are beneficial to human health. *Lactococcus lactis* 11/19-B1 strain isolated from kiwi fruit stimulates innate immunity in silkworms. Intake of yogurt containing the living 11/19-B1 strain significantly decreases the level of low-density lipoproteins (LDLs) in high-LDL volunteers and improves atopic dermatitis in humans. In this study, the probiotic properties of the 11/19-B1 strain, such as sensitivity to antimicrobial compounds, biogenic amine production, some virulence genes for human health, antimicrobial activity, tolerance to gastric acid and bile acids, and ability to adhere to the intestinal mucosa, were evaluated. The 11/19-B1 strain did not show resistance to the tested antimicrobial compounds except cefoxitin and fosfomycin. In addition, no production of amines that can harm humans, the antimicrobial activity required for probiotics, and the absence of adhesion to Caco-2 cells suggest that it is unlikely to attach to the intestinal epithelium. The 11/19-B1 strain grew in 0.3% but not in 1% bile salt. In the presence of 2% skim milk, the survival rate of the 11/19-B1 strain under simulated gastrointestinal tract conditions was 67% even after 4 h. These results indicate that the 11/19-B1 strain may function as a probiotic or paraprobiotic to be utilized in the food industry.

## 1. Introduction

Lactic acid bacteria (LABs) are defined as bacteria that produce lactic acid by fermenting sugars, such as lactose and glucose, and have been found in many environments, such as animals and plants, fermented foods, and feces [1,2,3]. Live bacteria that are beneficial to the human body are called probiotics, and these have attracted particular attention from the viewpoint of health consciousness and preventive medicine in recent years [4]. In terms of human safety, only LABs isolated from the human intestinal tract are recommended for human use as probiotics. However, in recent years, it has been reported that more beneficial probiotics have been isolated from the environment and fermented foods. Probiotics need to reach the intestines alive, while a new group, termed paraprobiotics, defined as inactivated (non-viable) microbial cells with a beneficial effect on the human body even in killed bacteria, has been proposed [5].

*Lactococcus lactis* is a Gram-positive cocci LAB that is a known probiotic and used in the production of many fermented foods such as yogurts and cheese, and some *L. lactis* strains produce various bacteriocins such as nisin, lactococcin, and diacetin [6,7,8]. Many in vitro and in vivo studies on the properties of probiotics and/or paraprobiotics of *L. lactis* isolated from the environment and food have been reported, and it has been revealed that *L. lactis* modulates host immunity [2]. Therefore, *L. lactis* is effective in strengthening the host’s immunity and improving immune-related symptoms. On the other hand, certain strains of *L. lactis* produce nisin, which has an antimicrobial effect. Due to its antimicrobial activity, nisin is used as a preservative food additive in more than 50 countries. In this way, it is important to note that although *L. lactis* is effective as a probiotic and/or paraprobiotic, it is strain-specific and not common to all *L. lactis*. Therefore, it is important to examine the characteristics of each strain in detail.

In silkworms, the living *L. lactis* 11/19-B1 strain isolated from kiwi fruit markedly stimulates innate immunity and shows certain differences in sugar utilization and enzymatic characteristics in comparison with other *L. lactis* strains [9]. Furthermore, we reported that the intake of yogurt containing the living 11/19-B1 strain significantly decreased the level of low-density lipoproteins (LDLs) in high-LDL volunteers compared to the control yogurt [10]. Moreover, the living 11/19-B1 strain improved atopic dermatitis (AD) in humans, and the killed 11/19-B1 strain improved AD-related dorsal skin and ear lesions in a 1-fluoro-2,4-dinitrobenzen-induced AD mouse model [11]. 

Probiotics must be generally regarded as safe (GRAS) for application in animal and human systems. The suitable properties of probiotics are: (1) antimicrobial activity to suppress the growth of other bacteria; (2) no virulence factors against the host; (3) no ability to produce amines; (4) sensitivity to antimicrobials; (5) tolerance to gastric acid and bile acids; and (6) ability to adhere to the intestinal mucosa. In this study, we evaluated the probiotic properties of the *L. lactis* 11/19-B1 strain.

## 2. Materials and Methods

### 2.1. Bacterial Strains

Microorganisms used in this study consisted of three strains of *L. lactis* [11/19-B1 (11/19-B1), JCM5805, and NBRC12007], *Lactobacillus delbrueckii* subsp. *Bulgaricus* LB-12 strain (*L. bulgaricus*), *Bifidobacterium lactis* Bb-12 strain (*B. lactis*), and our laboratory stocked two strains of genus *Pediococcus* (P1 and P2 strain) isolated from fermented soy paste (miso). The *L. lactis* JCM5805 and NBRC12007 strains were purchased from Riken BioResource Research Center (Tsukuba, Ibaraki, Japan), and the *L. lactis* 11/19-B1, the *L. bulgaricus* LB-12, and the *B. lactis* Bb-12 strains were kindly provided by Rakuo Kyodo Milk Co., Ltd. (Motomiya, Fukushima, Japan). The *L. lactis* JCM5805 and NBRC12007, *L. bulgaricus,* and *B. lactis* strains were used as reference controls to evaluate the probiotic properties of the 11/19-B1 strain. The *L. lactis* 11/19-B1, *L. bulgaricus,* and *B. lactis* strains are contained in the previously reported yogurt [10,11]. The *L. lactis* JCM5805 strain activates plasmacytoid dendritic cells and induces type-1 interferon [12]. The *L. lactis* NBRC12007 is a nisin A-producing strain [13]. For testing the antimicrobial activity of the LABs, *Staphylococcus aureus* (NBRC12732), *Pseudomonas aeruginosa* (NBRC12689), *Klebsiella pneumoniae* (NBRC14940), and *Escherichia coli* (NBRC3972) were used as indicator bacteria, with all strains purchased from Riken. For testing biogenic amine production, *Enterococcus faecalis* (JCM5803) purchased from Riken and *E. coli* (NBRC3972) were used as controls. All strains were stored at −80 °C using Microbank (Pro-Lab Diagnostics, Richmond Hill, ON, Canada) until use in the various tests. 

### 2.2. Sentitivity to Antibiotics

The sensitivity to antibiotics of each LAB used in this study was basically assessed with CLSI-M45 standard with some modifications [14]. Frozen-stocked bacteria, except *B. lactis*, were reactivated on De Man, Rogosa, and Sharpe (MRS) medium (Becton, Dickinson, and Company, Franklin Lakes, New Jersey, USA) containing agar with a final concentration of 1.5% at 35 °C under 5% CO_2_ for 24 h. *B. lactis* was reactivated on a MRS agar plate at 35 °C under anaerobic conditions for 24 h. Reactivated bacteria were subcultured once under the same conditions. Tested bacteria were selected and suspended in sterilized saline to the 0.5 McFarland turbidity standard equivalent. To prepare the samples for the antibiotic sensitivity test using a DP32 or DP43 dry plate (Eiken chemical Co., Tokyo, Japan), 25 µL of 11/19-B1 and JCM5805 suspension at 0.5 McFarland were added to 12 mL of MHBII (Becton, Dickinson, and Company) containing lysed horse blood (Nippon BIO-TEST laboratories, Asaka, Saitama, Japan) at a final concentration of 2.5%. On the other hand, 25 µL of *L. bulgaricus* suspension was added to 12 mL of MHBII containing lysed horse blood at a final concentration of 5%. Furthermore, 100 µL of prepared samples was added into wells of the DP32 or DP43 dry plate, and the plates were incubated for 24 h at 35 °C under aerobic conditions for the 11/19-B1 and JCM5805 strains or under 5% CO_2_ for the *L. bulgaricus* LB-12 strain.

### 2.3. Production of Amines

To detect biogenic amine (histamine, tyramine, putrescine, and cadaverine) production by the 11/19-B1 strain in culture medium, we carried out an amino acid decarboxylase test. Before detection, the 11/19-B1 was subcultured seven times in MRS broth with 0.1% (*w*/*v*) glucose, 0.005% pyridoxal 5-phosphate monohydrate (Tokyo Chemical Industry Co., Tokyo, Japan), and 0.1% (*w*/*v*) of each precursor amino acid: histidine hydrochloride (Themo Fischer Scientific, Waltham, MA, USA) for histamine, tyrosine (FUJIFILM Wako Chemicals Co., Osaka, Japan) for tyramine, ornithine hydrochloride (FUJIFILM Wako Chemicals Co., Osaka, Japan) for putrescine, and lysine monohydrochloride (Sigma-Aldrich Co., Saint Louis, MO, USA) for cadaverine [15]. A positive result was determined by the color change of bromocresol purple (Tokyo Chemical Industry) from yellow to purple.

### 2.4. Presence of Virulence Genes

We reported the total genome sequence and predicted proteins of 11/19-B1 (Genbank accession No. BLYG01000001.1 to BLYG01000006.1) previously [16]. In this study, we investigated whether 11/19-B1 possesses virulence genes such as *agg* (aggregation substance), *gelE* (gelatinase), *esp* (enterococcal surface protein), *efaAfs* and *efaAfm* (cell wall adhesins), and *cylA*, *cylB*, *cylM*, *cylLL*, and *cylLS* (cytolytic activity).

### 2.5. Antibacterial Activity

To detect the antibacterial activity against indicator bacteria (*S. aureus*, *P. aeruginosa*, *K. pneumoniae,* and *E. coli*), a spot-on-the-lawn assay was carried out with some modification [17]. Briefly, each indicator bacterium was incubated in 5 mL brain heart infusion (BHI) broth (Nissui Pharmaceutical Co., Tokyo, Japan) at 35 °C overnight. Moreover, 50 µL of the culture was added to 5 mL of BHI soft agar (0.7% agar), and this mixture was poured onto BHI agar. The tested LAB strains were grown in BHI broth for 48 h, and the supernatants were collected by centrifugation at 8000× *g* at 4 °C for 10 min. The collected supernatants were adjusted to pH 6.0 with NaOH, treated with or without heating at 80 °C for 10 min, filtrated using a 0.2-µm filter (KURABO industries, Osaka, Japan), and stocked at −30 °C until used. Moreover, 10 µL of the samples were then spotted onto BHI soft-agar plates containing the indicator bacterium. The plates were analyzed for bacterial growth inhibition after incubation at 35 °C for 18 h. The *L. lactis* NBRC12007 strain, a nisin-producing strain, was used as a positive control for the antibacterial activity test.

### 2.6. Suvival under Simulated Gastrointestinal Tract Conditions

The analysis of bacterial survival under simulated gastrointestinal tract conditions was performed according to the previous report [18]. The simulated gastric juice was prepared as follows: 0.06 g of FaSSGF (Boirelevant Com., London, United Kingdom) was dissolved in 1000 mL of 0.2% saline (adjusted to pH 1.6 with HCl), and then pepsin (Sigma-Aldrich Co., Saint Louis, MO, USA) was added at a final concentration of 0.1 mg/mL. This juice was filtrated before use. The tested bacteria, the 11/19-B1 and *B. lactis* Bb-12 strains, were incubated in 10 mL of MRS broth at 35 °C for 24 h under aerobic or anaerobic conditions, respectively. After centrifugation at 6800× *g* for 10 min at 4 °C, the resultant pellet from each culture was washed twice with phosphate-buffered saline (PBS) and then resuspended in 0.5% saline. A 200 μL aliquot of the cell suspension was mixed with 300 μL of saline or 10% skim milk and 1 mL of simulated gastric juice. After incubation at 37 °C for 0, 60, 120, 180, and 240 min, the number of viable cells was counted by culturing on MRS agar at 35 °C after 72 h to evaluate tolerance against simulated gastric juice. 

### 2.7. Susceptibility to Bile Salts

The susceptibility of the LABs used in this study to NaCl, pH, and bile salts was analyzed according to the method of Drosinos et al. [19], with some modifications. Tested LABs were grown on MRS agar at 35 °C under 5% CO_2_ for 24 h and subsequently subcultured in MRS broth at 35 °C under 5% CO_2_ for 24 h. Moreover, 50 µL of the cultures was added to 5 mL of modified MRS broth, with the addition of 4.5% or 6% of NaCl and 0.3% or 1% (*v*/*v*) of bovine bile, adjusted to pH 2.0 with HCl, and incubated at 35 °C for 3 days. The growth of bacteria strains in the medium was demonstrated by visual turbidity, and the replication ability of the bacteria was examined under different conditions. 

### 2.8. Adhesion Assay to the Human Colon Adenocarcinoma Cell Line Caco2

Adhesion assays were performed according to the method described by Guglielmotti et al. [20], with some modifications. The human colon adenocarcinoma cell line Caco-2 (Riken BioResource Research Center RCB0988) was used in this study. The cells were grown in Dulbecco’s Modified Eagle Essential Medium (Nissui Pharmaceutical Co., Tokyo, Japan) supplemented with 20% fetal calf serum and L-glutamine. The frozen, stocked 11/19-B1 cells were plated onto MRS agar and cultivated aerobically at 35 °C for 24 h. A few colonies of bacteria were cultured in 10 mL of MRS broth at 37 °C for 22 h under aerobic conditions. Moreover, 100 µL of 10 mL culture were inoculated into fresh MRS broth and cultured at 37 °C for 22 h. The cultures were centrifuged at 4000 rpm. After removing the supernatant, the pellets were resuspended in 2 mL of PBS. The 2 × 10^5^ Caco-2 cells were seeded per well of 24-well microplates and cultured in the fresh media mentioned above until 90% confluent. Three wells of Caco-2 monolayers were inoculated with approximately 7.2 × 10^7^ of the 11/19-B1 cells/well, and the plates were incubated at 35 °C under microaerophilic conditions. After 1 h of incubation, the monolayers were washed carefully three times with PBS, and the Caco-2 cells were lysed with 1 mL of ice-cooled 0.2% (*v*/*v*) Triton-X 100 solution (Sigma-Aldrich Co., Saint Louis, MO, USA) and vigorous pipetting. The Caco-2 cell lysates and the 11/19-B1 strain were collected and serially diluted in saline, and then the 11/19-B1 strain was plated on MRS agar and incubated at 35 °C for 72 h. All viable colonies were counted, and the CFU (colony-forming unit)/mL was calculated. Adhesion values (%) were calculated as follows: % Adhesion = (V1/V0) × 100, where V0 is the viable count of bacteria added initially and V1 is the viable bacteria count that adhered to Caco-2 cells at the end of the experiment. Two independent assays were carried out. 

### 2.9. Statistical Analysis

The survival of each LAB under simulated gastrointestinal tract conditions was determined in three independent trials. The average of CFU/mL and the standard deviation of each LAB were calculated.

## 3. Results and Discussion

### 3.1. Sentitivity to Antibacterial Compounds

Probiotics included in foods should not confer antimicrobial-resistant genes on other gut microbiota [21]. Minimum inhibitory concentrations (MICs) of antibacterial compounds against three LABs, the *L. lactis* 11/19-B1 and JCM5805, and *L. bulgaricus* LB-12 strains, were determined. The test was repeated two or three times, and the highest MIC values are shown in Table 1. The JCM5805 strain was used for comparison with 11/19-B1. The FEEDAP Panel (Table 2 in the 2018 EFSA [21]) establishes the microbiological cut-off values between susceptible and resistant for various LABs. The 11/19-B1, JCM5805, and *L. bulgaricus* LB-12 strains were inhibited at lower concentrations than the established cut-off values, at least for ampicillin, clindamycin, gentamicin, vancomycin, and erythromycin, with all strains showing resistance to cefoxitin and fosfomycin. The 11/19-B1 strain showed an equal to slightly higher MIC compared to the JCM5805 strain for all tested antibiotics. These results demonstrated that 11/19-B1 did not confer antimicrobial resistance genes on other microbiota and may be used as a food additive. 

Probiotics need to be safe for humans, but there have been cases of infection in association with some LABs, and there is a debate about safety and risk [22,23]. *L. lactis* does not adhere to the intestinal tract and, except for some cases, is considered safe, although it has been reported to cause bacteremia and endocarditis [24]. As shown in Table 1, the 11/19-B1 strain was inhibited at lower concentrations than the established cut-off values, at least for ampicillin, clindamycin, gentamicin, vancomycin, and erythromycin. This indicates that even if an infection occurs due to the 11/19-B1 strain, it can be treated with antibiotics.

### 3.2. Production of Amines and the Presence of Virulence Genes

Dairy products are made by fermenting milk with LABs, known as starter strains. LABs are known to produce biogenic amines (BAs), which are derived from various amino acids. It was reported that some BAs induce toxic effects such as headaches, heart palpitations, vomiting, diarrhea, and hypertensive crises [25]. It has been reported that dairy products contain large amounts of BAs, such as tyramine, histamine, putrescine, 2-phenylethylamine, etc., which are associated with these symptoms [26,27]. Therefore, BAs produced by the 11/19-B1 strain were investigated.

*E. coli* and two strains of the genus *Pediococcus* (P1 and P2) produced cadaverine and histamine, respectively, while the 11/19-B1 strain, like the control strain *E. faecalis* JCM5803, did not produce histamine, putrescine, tyramine, or cadaverine (Figure 1). Some *Lactococcus* species produce histamine, putrescine, or 2-phenyl-ethylamine [25]. The production of 2-phenyl-ethylamine by the 11/19-B1 strain is unknown, and the 11/19-B1 strain, unlike other *L. lactis* strains, does not produce tyramine or putrescine. Histamine and tyramine are known as particularly dangerous BAs as they may cause more severe symptoms [28,29]. Upon consumption of large amounts of cheese, the tyramine contained in the cheese can cause symptoms such as diet-induced migraines, increased cardiac output, nausea, vomiting, respiratory disorders, and elevated blood glucose [28,29]. High-level histamine intake can also result in harmful physiological effects due to its psychoactive and vasoactive properties, resulting in potentially life-threatening effects [30,31,32,33]. 

Indole, a metabolite of tryptophane, is synthesized by the tryptophanase of some intestinal bacteria, such as *Escherichia coli*. Once indole is absorbed from the intestinal tract, it accumulates in the liver through the portal vein, where it is converted to indoxyl by oxidative metabolism. Thereafter, indoxyl becomes indoxyl sulfate through the action of sulfotransferase 1A1 (SULT1A1). Indoxyl sulfate acts as a cardiotoxin and uremic toxin, and although it is excreted from the body with urine when renal function is functioning normally, it accumulates in the blood due to a decrease in urine volume due to a decrease in renal function. Indole and indoxyl sulfate are considered to be factors in the progression of renal failure [34]. The genome sequence information of the 11/19-B1 strain (Genbank accession No. BLYG01000001.1 to BLYG01000006.1) indicates that the present strain does not possess tryptophanase and estimates that the present strain does not produce indole. It is thought that foods fermented with the 11/19B1 strain do not contain indole, and patients with chronic kidney disease can consume these foods. The selection of specific starter strains that lack the accumulation pathway of these harmful BAs is essential for obtaining high-quality foods with a reduced BA content, and the 11/19-B1 strain could be suitable for use as a starter strain.

The genome and predicted protein sequence data of the 11/19-B1 strain (Genbank accession No. BLYG01000001.1 to BLYG01000006.1) showed no virulence genes such as *agg* (aggregation substance), *gel*E (gelatinase), *esp* (enterococcal surface protein), *efa*Afs and *efa*Afm (cell wall adhesins), or *cyl*A, *cyl*B, *cyl*M, *cyl*L_L_, and *cyl*L_S_ (cytolytic activity). These genes are recognized as virulence genes of the genus Enterococcus [35]. These findings suggested that the 11/19-B1 strain would not cause harm to consumers.

### 3.3. Antibacterial Activity

It has been reported that several LABs produce bacteriocin to prevent the growth of pathogenic microbes such as *Listeria monocytogenes*, *Bacillus cereus*, or *Staphylococcus aureus* [36,37,38,39,40]. Nisin A, produced by certain strains of *L. lactis*, is a bacteriocin consisting of 34 amino acids that shows an antimicrobial effect against Gram-positive bacilli such as the genera *Bacillus* and *Clostridium* [41]. This bacteriocin is also thought to have very little effect on the intestinal flora in the lower intestinal tract, as it is inactivated by pancreatin, an intestinal enzyme [42]. Furthermore, it was recognized as GRAS (Generally Recognized As Safe) by the FDA (Food and Drug Administration) in 1988 as a preservative food additive in more than 50 countries, including Japan. The control strain NBRC12007, a nisin A-producing strain, showed antimicrobial activity against only *S. aureus* among the strains tested in this study. The 11/19-B1 strain was shown to possess lactococcin as a bacteriocin based on the genome sequence; however, this strain did not show antibacterial activity against Gram-positive cocci such as *S. aureus* and Gram-negative bacilli such as *E. coli*, *K. pneumoniae*, and *P. aeruginosa* in this study. Such antibacterial properties of the 11/19-B1 strain may be advantageous for foods produced by the fermentation of multiple lactic acid bacteria, such as yogurt. Furthermore, even if the 11/19-B1 strain reaches the intestines alive, its bacteriocin products, such as lactococcin, are thought to have little effect on the intestinal flora.

### 3.4. Suvival under Simulated Gastrointestinal Tract Conditions

To examine whether the 11/19-B1 strain reaches the intestine alive, we examined its survival under various conditions (simulated gastric tract conditions, low pH, NaCl, and bile salts). In the fasted state, the pH in the stomach of a healthy person is about 1.7 (1–2) [43]. In addition, bacteria must be able to withstand pH 2.5 to 3.5 to survive in the stomach [44]. As shown in Table 2, none of the LABs in this experiment showed growth at pH 2. 

The digestion of foods is known to take 2–4 h, so the viability of each LAB was examined in simulated gastric juice up to 4 h after exposure. Under simulated gastric tract conditions, the viable bacteria numbers of 11/19-B1 and *B. lactis* decreased from 2.3 × 10^8^ to 0 and from 3.2 × 10^7^ to 0 within 2 h, respectively. In addition, JCM5805 was reduced from 8.4 × 10^7^ to 0 within 1 h. As shown in Figure 2, the survival rate of these bacteria was improved by the presence of skim milk at a final concentration of 2%, with the survival rate of 11/19-B1 reaching 67% even after 4 h (Figure 2). The pH in the stomach rises to 4.5 after eating, but the 11/19-B1 strain was viable in the presence of skim milk, even in simulated gastric juice with pH 1. These findings suggest that the 11/19-B1 strain in yogurt can survive, reach the intestine, and work as a probiotic, even when ingested on an empty stomach.

Along with pH tolerance, bile acid tolerance is an important factor for probiotics that reach the intestine alive. Furthermore, salt tolerance for survival in foods with high salt concentrations is also an important factor. As shown in Table 2, the 11/19-B1 strain showed the same salt tolerance as the other LABs tested in this study. *B. lactis* showed tolerance to 1% bile salt, as previously reported. Other LABs, including 11/19-B1, grew in 0.3% but not in 1% bile salt. 

### 3.5. Ability to Adhere to Caco-2 Cells, Human Colon Adenocarcinoma Cells

LABs can influence the adhesion of specific pathogenic microbes by competing for binding sites on intestinal epithelial cells, thereby preventing the development of infections [45,46]. Therefore, the adhesion of LAB to the gastrointestinal epithelium is one of the factors that determines whether it functions as a probiotic. To investigate whether the 11/19-B1 strain can adhere to and settle in the colon, we undertook an adhesion assay using the human colon adenocarcinoma cell line, Caco-2 cells. The 11/19-B1 strain did not adhere alone. It was reported that the adhesion of LABs to intestinal epithelial cells is associated with the presence of mucin [47]. Goblet cells exist in the gastrointestinal epithelium of mammals, and a large amount of mucus (mucin) is produced from the goblet cells and coats the intestinal epithelium. Therefore, mucin is considered to be the main place of settlement for LABs. Caco2 cells are known to be incapable of mucin production, so it remains possible that the 11/19-B1 strain adheres to the intestinal epithelium along with other bacteria after binding with mucin.

### 3.6. The 11/19-B1 Strain as a Probiotic and/or Proparabiotic

Table 3 shows the in vivo probiotic and paraprobiotic effects of the 11/19-B1 strain and the mechanisms thought to be involved mechanisms, and Table 4 shows the in vitro probiotic properties of this study.

Consumption of yogurt containing live 11/19-B1 strains significantly reduced LDL levels, lowered SCORAD scores in human AD patients, and activated innate immunity in silkworms [9,10,11]. In addition, in a mouse AD model using DNFB, it was reported that ingestion of the heat-killed 11/19-B1 strain improved AD-related dorsal skin and ear lesions [11].

It is not clear whether the hypocholesterolemic effect is related to changes in the gut microbiota or whether it is a direct function of the strain. In this study, the 11/19-B1 strain did not show anti-microbial activity or adhesion to Caco-2 cells. These results suggest that the hypocholesterolemic effect is a direct function of the strain rather than a result of alterations in the gut microbiota. On the other hand, *L. lactis* has also been reported to modify the gut microbiota, even when heat-killed bacteria were used [48,49]. It is possible that compounds produced by the 11/19-B1 strain and/or bacterial components contained in the yogurt altered the intestinal microbiota as a prebiotic and/or paraprobiotic, resulting in a hypocholesterolemic effect. 

The mechanism for lowering the SCORAD score of human AD patients by ingestion of 11/19-B1 yogurt remains unclear, but the heat-killed 11/19-B1 strain causes the activation of innate immunity in silkworms and the improvement in AD-related dorsal skin and ear lesions in mouse AD models. These phenomena are thought to be due to the paraprobiotic effects of the 11/19-B1 strain. Previous studies have shown that LABs, whether alive or dead, have an immunoregulatory effect on hosts [50,51]. As with the 11/19-B1 strain, another *L. lactis* strain, JCM20101, also improved AD-related dorsal skin and ear lesions in the mouse AD model [11]. This effect may be a common feature of *L. lactis* strains. The reason for activation of immunity with the 11/19-B1 strain may be due to genomic DNA or cell wall components. Therefore, the 11/19-B1 strain can be used as a paraprobiotic through its application as an additive not only in yogurt but also as an additive in various foods without the risk of infection.

## 4. Conclusions

In this study, it was suggested that strain 11/19-B1 may reach the intestines alive when ingested as a food additive in products such as yogurt. In addition, since it has been shown to be sensitive to many antibiotics, the risk of it bestowing drug resistance on other intestinal bacteria upon reaching the intestine alive is low. As no production of bacteriocin was observed, even when used as a food additive, this strain does not prevent the growth of bacteria beneficial to humans, and the risk of dysbiosis in the intestine is also considered to be low. In addition, as the ingestion of sterilized 11/19-B1 improved atopic dermatitis in a 1-fluoro-2, 4-dinitrobenzen-induced atopic dermatitis mouse model, as described above, 11/19-B1 may function as a paraprobiotic. It has been reported that certain killed *L. lactis* bacteria change the intestinal flora and suppress aging. We have been investigating whether the intake of strain 11/19-B1 affects fecal indole, blood indoxyl sulfate, the improvement of medical conditions, and changes in the gut microbiota in patients undergoing dialysis due to renal failure. Since this study targeted dialysis patients, it would be possible to investigate the risks of ingesting the 11/19-B1 strain. Furthermore, using a mouse model of age-related hearing loss, we are investigating whether ingestion of killed bacteria from the 11/19-B1 strain will inhibit the progression of age-related hearing loss. Yoghurts containing live bacteria are already on the market, but it would be possible to use this killed bacterium as a paraprobiotic in a variety of foods.

## Figures and Tables

**Figure 1 microorganisms-11-02949-f001:**
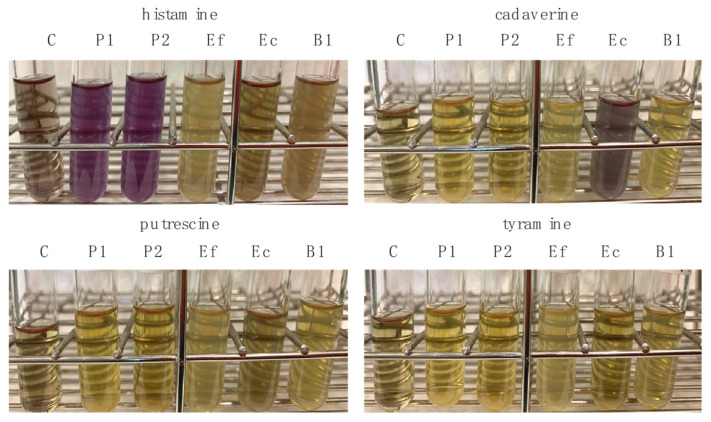
Biogenic amine production of the *L. lactis* 11/19-B1 strain and other LABs for comparison. The *L. lactis* 11/19-B1 strain and *E. faecalis* did not produce any of the biogenic amines tested in this study. C: negative control; P1 and P2: genus *Pediococcus* isolated from miso; Ef: *E. faecalis*; Ec: *E. coli*; B1: *L. lactis* 11/19-B1.

**Figure 2 microorganisms-11-02949-f002:**
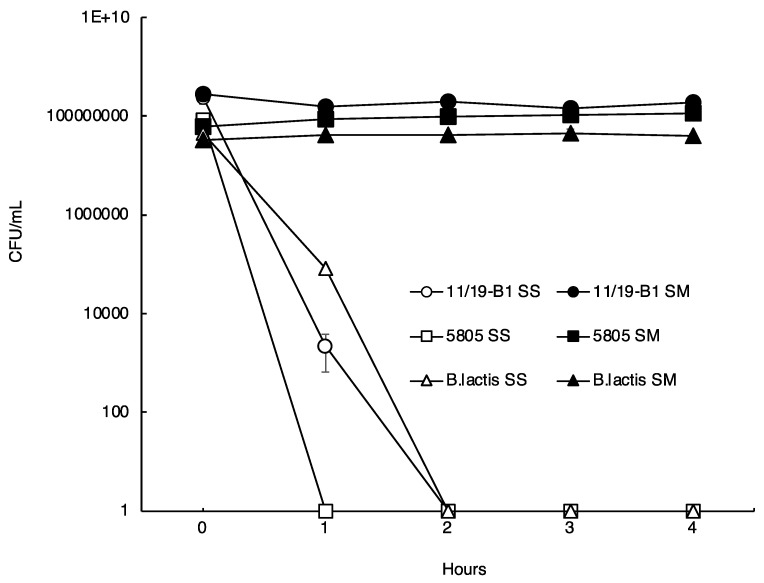
Survival curves of the *L. lactis* 11/19-B1 strain and other LABs for comparison under simulated gastrointestinal tract conditions. Bars indicate the standard deviation. Circles: *L. lactis* 11/19-B1; squares: *L. lactis* JCM5805; triangles: *B. lactis* Bb-12; open symbols: SS (in saline); filed symbols: SM (in skim milk).

**Table 1 microorganisms-11-02949-t001:** Minimum inhibitory concentrations (MICs; µg/mL) of antibacterial compounds against each LAB strain in this study.

	*L. lactis*		*L. bulgaricus*
	11/19-B1	JCM5805	LB-12
ampicillin	0.5	≤0.12	0.25
cefmetazole	32	16	32
flomoxef	8	4	8
imipenem	≤0.25	≤0.25	≤0.25
minocycline	≤2	≤2	≤2
clindamycine	0.12	≤0.06	0.25
levofloxacin	1	2	2
gentamicin	0.5	≤0.25	1
vancomycin	≤0.5	≤0.5	≤0.5
oxacillin	1	0.5	1
cefazolin	4	2	2
arbekacin	1	≤0.25	2
cefoxitin	>16	>16	>16
erythromycin	≤0.12	≤0.12	≤0.12
teicoplanin	≤0.5	≤0.5	≤0.5
linezolid	4	1	4
fosfomycin	128	64	128
sulfamethoxazole/trimethoprim	>38/2	≤9.5/0.5	≤9.5/0.5

**Table 2 microorganisms-11-02949-t002:** Tolerance to pH, NaCl, and bile salt of *L. lactis* 11/19-B1 and other tested LABs.

		pH 2	NaCl		Bile Salt	
			4.5%	6%	0.3%	1%
*L. lactis*	11/19-B1	−	+	+	+	−
	JCM5805	−	+	+	+ *	−
*L. bulgaricus*	LB-12	−	+	+	+	−
*B. lactis*	Bb-12	−	+	+	+	+ **

+: presence of growth; −: absence of growth Testing was carried out in duplicate at one time. * indicates that one of two trials was grown. ** indicates lower growth than that in 0.3% bile salt.

**Table 3 microorganisms-11-02949-t003:** Probiotic and paraprobiotic effects of the *L. lactis* 11/19-B1 strain.

	Object	Effect	Mechanisms	Reference No.
probiotics	Silkworm	antimicrobial effect	activates the innate immunity	[9]
(viable)	Human	decreases LDL level	unknown	[10]
		improves atopic dermatitis	unknown	[11]
paraprobiotics	Silkworm	antimicrobial effect	activates the innate immunity	[9]
(heat killed)	Mouse	improves AD-related dorsal skin and ear lesions	activates Treg and decreases Th1, Th2, and Th17 cells	[11]

**Table 4 microorganisms-11-02949-t004:** Probiotic properties of the L. lactis 11/19-B1 strain from this study.

Sensitivity to Antimicrobial Compounds	
ampicillin	sensitive
clindamycin	sensitive
gentamicin	sensitive
vancomycin	sensitive
erythromycin	sensitive
cefoxitin and fosfomycin	naturally resistant
Production of biogenic amine	
histamine, cadaverine, putrecine, and tyramine	none
indole (tryptophanase gene)	none (from genome information)
Virulence gene	
*agg* (aggregation substance)	none (from genome information)
*gelE* (gelatinase)	none (from genome information)
*esp* (enterococcal surface protein)	none (from genome information)
*efaAfs* and *efaAfm* (cell wall adhesins)	none (from genome information)
*cylA*, *cylB*, *cylM*, *cylLL*, and *cylLS* (cytolytic activity)	none (from genome information)
Antibacterial activity to	
*S. aureus*	none
*P. aeruginosa*	none
*K. pneumoniae*	none
*E. coli*	none
Survival under simulated gastrointestinal tract conditions	
after 4 h of incubation	
in saline	0% survive
in skim milk	64% survive
Tolerance to pH 2	growth
Tolerance to NaCl	
4.5%	growth
6%	growth
Tolerance to bile salt	
0.3%	growth
1%	absence of growth
Ability to adhere to Caco-2 cells	none

Genome information: Genbank accession No. BLYG01000001.1 to BLYG01000006.1.

## Data Availability

The data presented in this study are available on request from the corresponding author.

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
