# Peer review of "Characterization of Lactococcus lactis 11/19-B1 Isolated from Kiwi Fruit as a Potential Probiotic and Paraprobiotic"

_microorganisms, 2023, doi:10.3390/microorganisms11122949_

Round 1

Reviewer 1 Report

Comments and Suggestions for Authors

Probiotics are live bacteria used as food additives and are beneficial to human health.The authors isolated a 11 / 19-B1 strain from kiwifruit that stimulate innate immunity in worms.The probiotic properties of the 11 / 19-B1 strain were evaluated, such as susceptibility to antimicrobial compounds, biological amine production, virulence factors, anti-microbial activity, tolerance to gastric and bile acids, and the ability to adhere to the intestinal mucosa.

But there are problems with the manuscript as follows:

The format of the contents in Table 1 needs to be centered.

The picture in Figure 1 is not very clear.

The color of Figure 2 is black and white, and the distinction is not very obvious.

Lactococcus lactis 11 / 19-B1 bacteria cannot exclude the risk of infection, even if it can be treated, but there will be some controversy for food fermentation, which is not further explored in this paper.

Why did not set the sensitivity to bile salt, but directly choose 0.3% and 1.0%?

There are few pictures and tables in the whole article, which seems that the test results are not convincing enough. The table pictures can be appropriately added to improve the persuasiveness of the article

Comments on the Quality of English Language

English quality suitable for publication

Author Response

Probiotics are live bacteria used as food additives and are beneficial to human health. The authors isolated a 11 / 19-B1 strain from kiwifruit that stimulate innate immunity in worms. The probiotic properties of the 11 / 19-B1 strain were evaluated, such as susceptibility to antimicrobial compounds, biological amine production, virulence factors, anti-microbial activity, tolerance to gastric and bile acids, and the ability to adhere to the intestinal mucosa.

But there are problems with the manuscript as follows:

Comment 1

The format of the contents in Table 1 needs to be centered.

Thank you for your suggestion.

The contents in Table 1 were revised to be centered.

Comment 2

The picture in Figure 1 is not very clear.

Thank you for your suggestion.

Figure 1 was revised to high resolution photograph.

Comment 3

The color of Figure 2 is black and white, and the distinction is not very obvious.

Thank you for your suggestion.

We revised the Figure legends. A sentence “open symbols: SS (in saline), filed symbols: SM (in skim milk) was added in line 296.

Comment 4

Lactococcus lactis 11 / 19-B1 bacteria cannot exclude the risk of infection, even if it can be treated, but there will be some controversy for food fermentation, which is not further explored in this paper.

Thank you for your comment.

L. lactis is a safe microbe as a food additive certified by GRAS, and patients infected with L. lactis have underlying medical conditions. The risk of infection in these patients needs to be examined in the future. In the case of 11/19-B1 strain, yogurt fermented with present strain as a starter is already commercially available, and no health hazards have been reported when high LDL volunteers, atopic dermatitis patients, and healthy volunteers ingest the yogurt (ref No. 10, 11). In addition, dialysis patients are currently consuming the same yogurt, but there have been no reports of health hazards in the same way. However, as long as there is a risk of infection, we think this paper has shown that antibiotic treatment is effective in infection with present strain.

Comment 5

Why did not set the sensitivity to bile salt, but directly choose 0.3% and 1.0%?

Thank you for your comment.

Goldin and Gorbach (1992) reported that 0.15 to 0.3% is the appropriate concentration for screening for bile acid tolerance of probiotics used in humans. For this reason, we selected 0.3% as a low concentration of bile salt. Bifidobacterium lactis Bb-12 strain, control strain in this study, was reported to be moderately tolerance at bile acid concentrations of 1%, and many researchers choose 0.3% and 1.0% directly for bile salt tolerance test.

Comment 6

There are few pictures and tables in the whole article, which seems that the test results are not convincing enough. The table pictures can be appropriately added to improve the persuasiveness of the article.

As pointed out by the reviewer, we apologized for the few figures and tables.

The previously reported in vivo effects of the 11/19-B1 strain as probiotics and/or paraprobiotics are summarized in Table 3, and the in vitro properties as probiotics in this study are summarized in Table 4.

Reviewer 2 Report

Comments and Suggestions for Authors

The manuscript comprehensively characterizes Lactococcus lactis 11/19-B1 isolated from kiwi fruit as a potential probiotic and paraprobiotic. The authors have thoroughly investigated the strain's probiotic potential, including its antimicrobial activity, adhesion to intestinal cells, and ability to modulate the immune system. The manuscript is well-written and easy to follow.

Specific comments:

1- Introduction: The authors could provide more specific information about the potential applications of Lactococcus lactis as a probiotic or paraprobiotic.

2- Materials and Methods: The materials and methods section is well-written and detailed. 

3- Results: The authors could consider adding a table summarizing the key characteristics of Lactococcus lactis 11/19-B1.

4- Discussion: The authors should provide a more detailed discussion of the mechanism of action of Lactococcus lactis 11/19-B1 as a probiotic or paraprobiotic. For example, they could discuss how the strain's antimicrobial activity, adhesion to intestinal cells, and ability to modulate the immune system contribute to its probiotic or paraprobiotic effects. Also, they could discuss how the 11/19-B1 strain compares to other strains in terms of its antimicrobial activity, tolerance to gastric acid and bile acids, and ability to adhere to intestinal cells.

5- Conclusion: The authors should discuss the future directions of their research. For example, they could discuss the need to conduct in vivo studies to confirm the probiotic or paraprobiotic effects of the 11/19-B1 strain. They could also discuss the need to develop a food product that contains the strain.

Comments on the Quality of English Language

Minor editing of English language required

Author Response

The manuscript comprehensively characterizes Lactococcus lactis 11/19-B1 isolated from kiwi fruit as a potential probiotic and paraprobiotic. The authors have thoroughly investigated the strain's probiotic potential, including its antimicrobial activity, adhesion to intestinal cells, and ability to modulate the immune system. The manuscript is well-written and easy to follow.

We appreciate for your review and comments.

Specific comments:

1- Introduction: The authors could provide more specific information about the potential applications of Lactococcus lactis as a probiotic or paraprobiotic.

Thank you very much for your comments.

Authors added the information of applications of L. lactis as a probiotic or paraprobiotic in line39 to 48.

2- Materials and Methods: The materials and methods section is well-written and detailed.

We appreciated for your kind comments.

3- Results: The authors could consider adding a table summarizing the key characteristics of Lactococcus lactis 11/19-B1.

Thank you very much for your suggestions.

The previously reported in vivo effects of the 11/19-B1 strain as probiotics and/or paraprobiotics are summarized in Table 3, and the in vitro properties as probiotics in this study are summarized in Table 4.

4- Discussion: The authors should provide a more detailed discussion of the mechanism of action of Lactococcus lactis 11/19-B1 as a probiotic or paraprobiotic. For example, they could discuss how the strain's antimicrobial activity, adhesion to intestinal cells, and ability to modulate the immune system contribute to its probiotic or paraprobiotic effects. Also, they could discuss how the 11/19-B1 strain compares to other strains in terms of its antimicrobial activity, tolerance to gastric acid and bile acids, and ability to adhere to intestinal cells.

Thank you very much for your suggestion.

The 11/19-B1 strain did not show antimicrobial activity against the bacteria used in this study. We mentioned its advantages in line 271 to 275. On the other hand, unpublished data showed that the supernatant of present strain culture without pH adjustment is showed antimicrobial activity against S. aureus. This suggests that when making yogurt fermented using the 11/19-B1 strain, present strain inhibits the growth of S. aureus and production of its enterotoxin in food.

Probiotics are known to reduce the intestinal infection risk by adhesion to the intestinal epithelium, inhibiting the colonization of other bacteria, and this is mentioned in line 308 to 311. The 11/19-B1 strain did not adhere to Caco-2 cells which do not produce mucins in this study. Since mucin is known to be the key to adhere to intestinal cells, the possibility of 11/19-B1 strain adhere to intestinal epithelial cells was mentioned in line 314 to 320. In general, adhesion to mucin is required adhesin, but 11/19-B1 strain does not possess adhesin gene based on genome sequence. It has also been reported that LABs which adhere to intestinal epithelial cells are taken up by dendrotic cells and transported to the mammary gland, and that LABs ingested by infants may function as probiotics. These investigation has been in progress.

Although there is still insufficient data to discuss the immunomodulatory mechanism, Suzuki et al. reported that dead bacteria of the 11/19-B1 strain activated Tregs and suppressed Th1, Th2, and Th17 in mice (ref 11). It has been reported that suppression of this subset may improve AD pathology, and the 11/19-B1 strain is thought to improve AD-related dermatitis in mice by a similar mechanism. The reason for activation of immunity with the 11/19-B1strain may be due to genomic DNA or cell wall components.

This sentence is added in line 350 to 351.

5- Conclusion: The authors should discuss the future directions of their research. For example, they could discuss the need to conduct in vivo studies to confirm the probiotic or paraprobiotic effects of the 11/19-B1 strain. They could also discuss the need to develop a food product that contains the strain.

Thank you very much for your suggestion.

It has been reported that certain killed L. lactis bacteria change the intestinal flora and suppress aging. Indole, metabolites of tryptophane, is synthesized by tryptophanase of some intestinal bacteria such as Escherichia coli. Once indole is absorbed from the intestinal tract, it accumulates in the liver through the portal vein, where it converts to indoxyl by oxidative metabolism. Thereafter, indoxyl becomes indoxyl sulfate through the action of sulfotransferase 1A1 (SULT1A1). Indoxyl sulfate acts as a cardiotoxin and uremic toxin, and although it is excreted from the body with urine when renal function is functioning normally, it accumulates in the blood due to a decrease in urine volume due to a decrease in renal function. Indole and indoxyl sulfate are considered to be a factor in the progression of renal failure. The genome sequence information of 11/19-B1 strain (Genbank accession No. BLYG01000001.1 to BLYG01000006.1) indicates that the present strain does not possess tryptophanase and estimate that present strain does not produce indole. We have been investigating whether the intake of strain 11/19-B1 affects fecal indole, blood indoxyl sulfate, improvement of medical condition, and changes in the gut microbiota in patients undergoing dialysis due to renal failure. Since this study targeted dialysis patients, it would be possible to investigate the risks of ingesting the 11/19-B1 strain. Furthermore, using a mouse model of age-related hearing loss, we are investigating whether ingestion of killed bacteria from the 11/19-B1 strain will inhibit the progression of age-related hearing loss. Yoghurts containing live bacteria are already on the market, but it would be possible to use this killed bacterium as paraprobiotics in a variety of foods.

Manuscript was revised with above sentences in L226-L238 and L346-L353.

Reviewer 3 Report

Comments and Suggestions for Authors

Dear authors,

Please find attached the pdf file with my suggestions.

Is it important to perform the statistical analysis for the quantitative results.

Author Response

Please find attached the pdf file with my suggestions.

We found pdf file with your suggestion to our manuscript and appreciated for your review.

Is it important to perform the statistical analysis for the quantitative results.

As reviewer pointed out, authors revised figure legends in line 295.

Comment 1

Line 14

Please note that the results from this analysis were not provided.

Thank you for your suggestion.

The expression "virulence factor" was ambiguous, so it was revised to "some virulence genes to human health" in line 14.

Comment 2

Line 16-17

Please consider to rewrite this sentence to make clear what of these characteristics are desired for a probiotic strain.

Thank you for your suggestion.

We revised the sentence “Further, no production of the tested amines, antimicrobial activity against the tested bacteria or adhesion to CaCo-2 cells were observed“ to “In addition, no production of amines that can harm humans, and the antimicrobial activity required for probiotics, and the absence of adhesion to Caco-2 cells suggest that it is unlikely to attach to the intestinal epithelium” in line 17 -18.

Comment 3

Line 114 and 142

Please use microliters instead of the abbreviation in this case.

Thank you for your suggestion.

We revised “Fifty µl” to “50 µL” in line 124 and 152.

Comment 4

In Material and Methods section.

The methods used for cell adherence analysis are not provided.

Thank you for your suggestion.

The section “2.8. Adhesion assay to human colon adenocarcinoma cell line Caco2” was added in line 157 to 180.

Comment 5

In Material and Methods section.

The statistical analysis section is missed.

Thank you for your suggestion.

The section “2.9. Statistical Analysis” was added in line 181 to 184.

Comment 6

Line 150 and 162

Please consider to rewrite this sentence since the MIC is the antibiotics against the LABs.

Thank you for your suggestion.

We revised the sentences “Minimum inhibitory concentrations (MICs) of antibacterial compounds against three LABs, the L. lactis 11/19-B1 and JCM5805, and L bulgaricus LB-12 strains, were determined” in line 188 to 190 and “Minimum inhibitory concentrations (MICs; µg/mL) of antibacterial compounds against each LAB strain in this study” in line 200 to 201, respectively.

Comment 7

Line 181, 242, 243 and 251

Italic

Thank you for your suggestion.

The font of the scientific names that were pointed out has been revised to italic in line 294, 295, 304.

Reviewer 4 Report

Comments and Suggestions for Authors

Ishioka et al., characterized Lactococcus lactis strain 11/19-B1. The authors aimed to study the properties of probiotics: 1) antimicrobial activity to sup- 48 presses the growth of other bacteria, 2) no virulence factors against the host, 3) no ability 49 to produce amines, 4) sensitivity to antimicrobials, 5) tolerance to gastric acid and bile 50 acids, and 6) ability to adhere to the intestinal mucosa.

They performed the antimicrobial activity experiments to find out if the strain is able to suppress the growth of other bacteria and concluded “11/19-B1 did not confer antimicrobial resistance genes on other microbiota”. Secondly, 11/19-B1 strain demonstrated sensitivity to antimicrobials, therefore it can be treated with antibiotics. Thirdly, 11/19-B1 strain had no virulence genes, therefore being harmless to users. Interestingly, the authors demonstrated in simulated conditions that “the 11/19-B1 strain 238 in yogurt can survive, reach the intestine and work as a probiotic, even when ingested on 239 an empty stomach.”

The work is carefully planned, well-presented, and contains adequate amounts of results.

Author Response

They performed the antimicrobial activity experiments to find out if the strain is able to suppress the growth of other bacteria and concluded “11/19-B1 did not confer antimicrobial resistance genes on other microbiota”. Secondly, 11/19-B1 strain demonstrated sensitivity to antimicrobials, therefore it can be treated with antibiotics. Thirdly, 11/19-B1 strain had no virulence genes, therefore being harmless to users. Interestingly, the authors demonstrated in simulated conditions that “the 11/19-B1 strain 238 in yogurt can survive, reach the intestine and work as a probiotic, even when ingested on 239 an empty stomach.”

The work is carefully planned, well-presented, and contains adequate amounts of results.

We appreciated for your review and kind comments.